# Coresets for Clustering with Fairness Constraints

**Lingxiao Huang**[*]
Yale University, USA

**Shaofeng H.-C. Jiang**[*]
Weizmann Institute of Science, Israel

**Nisheeth K. Vishnoi**[*]
Yale University, USA

## Abstract

In a recent work, [20] studied the following "fair" variants of classical clustering problems such as $k$-means and $k$-median: given a set of $n$ data points in $\mathbb{R}^d$ and a binary type associated to each data point, the goal is to cluster the points while ensuring that the proportion of each type in each cluster is roughly the same as its underlying proportion. Subsequent work has focused on either extending this setting to when each data point has multiple, non-disjoint sensitive types such as race and gender [7], or to address the problem that the clustering algorithms in the above work do not scale well [42, 8, 6]. The main contribution of this paper is an approach to clustering with fairness constraints that involve *multiple, non-disjoint* types, that is *also scalable*. Our approach is based on novel constructions of coresets: for the $k$-median objective, we construct an $\varepsilon$-coreset of size $O(\Gamma k^2 \varepsilon^{-d})$ where $\Gamma$ is the number of distinct collections of groups that a point may belong to, and for the $k$-means objective, we show how to construct an $\varepsilon$-coreset of size $O(\Gamma k^3 \varepsilon^{-d-1})$. The former result is the first known coreset construction for the fair clustering problem with the $k$-median objective, and the latter result removes the dependence on the size of the full dataset as in [42] and generalizes it to multiple, non-disjoint types. Plugging our coresets into existing algorithms for fair clustering such as [6] results in the fastest algorithms for several cases. Empirically, we assess our approach over the **Adult**, **Bank**, **Diabetes** and **Athlete** dataset, and show that the coreset sizes are much smaller than the full dataset; applying coresets indeed accelerates the running time of computing the fair clustering objective while ensuring that the resulting objective difference is small. We also achieve a speed-up to recent fair clustering algorithms [6, 7] by incorporating our coreset construction.

## 1 Introduction

Clustering algorithms are widely used in automated decision-making tasks, e.g., unsupervised learning [43], feature engineering [33, 27], and recommendation systems [10, 40, 21]. With the increasing applications of clustering algorithms in human-centric contexts, there is a growing concern that, if left unchecked, they can lead to discriminatory outcomes for protected groups, e.g., females/black people. For instance, the proportion of a minority group assigned to some cluster can be far from its underlying proportion, even if clustering algorithms do not take the sensitive attribute into its decision making [20]. Such an outcome may, in turn, lead to unfair treatment of minority groups, e.g., women may receive proportionally fewer job recommendations with high salary [22, 38] due to their underrepresentation in the cluster of high salary recommendations.

To address this issue, Chierichetti et al. [20] recently proposed the fair clustering problem that requires the clustering assignment to be *balanced* with respect to a binary sensitive type, e.g., sex.[2] Given a set $X$ of $n$ data points in $\mathbb{R}^d$ and a binary type associated to each data point, the goal is to cluster the points such that the proportion of each type in each cluster is roughly the same as

---

[*]Authors are listed in alphabetical order of family names. Full version: [31].

[2]A type consists of several disjoint groups, e.g., the sex type consists of females and males.

its underlying proportion, while ensuring that the clustering objective is minimized. Subsequent work has focused on either extending this setting to when each data point has multiple, non-disjoint sensitive types [7] (Definition 2.3), or to address the problem that the clustering algorithms do not scale well [20, 41, 42, 8, 6].

Due to the large scale of datasets, several existing fair clustering algorithms have to take samples instead of using the full dataset, since their running time is at least quadratic in the input size [20, 41, 8, 7]. Very recently, Backurs et al. [6] propose a nearly linear approximation algorithm for fair $k$-median, but it only works for a binary type. It is still unknown whether there exists a scalable approximation algorithm for multiple sensitive types [6]. To improve the running time of fair clustering algorithms, a powerful technique called coreset was introduced. Roughly, a coreset for fair clustering is a small weighted point set, such that for any $k$-subset and any fairness constraint, the fair clustering objective computed over the coreset is approximately the same as that computed from the full dataset (Definition 2.1). Thus, a coreset can be used as a proxy for the full dataset – one can apply any fair clustering algorithm on the coreset, achieve a good approximate solution on the full dataset, and hope to speed up the algorithm. As mentioned in [6], using coresets can indeed accelerate the computation time and save storage space for fair clustering problems. Another benefit is that one may want to compare the clustering performance under different fairness constraints, and hence it may be more efficient to repeatedly use coresets. Currently, the only known result for coresets for fair clustering is by Schmidt et al. [42], who constructed an $\varepsilon$-coreset for fair $k$-means clustering. However, their coreset size includes a $\log n$ factor and only restricts to a sensitive type. Moreover, there is no known coreset construction for other commonly-used clusterings, e.g., fair $k$-median.

**Our contributions.** Our main contribution is an efficient construction of coresets for clustering with fairness constraints that involve multiple, non-disjoint types. Technically, we show efficient constructions of $\varepsilon$-coresets of size independent of $n$ for both fair $k$-median and fair $k$-means, summarized in Table 1. Let $\Gamma$ denote the number of distinct collections of groups that a point may belong to (see the first paragraph of Section 4 for the formal definition).

- Our coreset for fair $k$-median is of size $O(\Gamma k^2 \varepsilon^{-d})$ (Theorem 4.1), which is the first known coreset to the best of our knowledge.
- For fair $k$-means, our coreset is of size $O(\Gamma k^3 \varepsilon^{-d-1})$ (Theorem 4.2), which improves the result of [42] by an $\Theta(\frac{\log n}{\varepsilon k^2})$ factor and generalizes it to multiple, non-disjoint types.
- As mentioned in [6], applying coresets can accelerate the running time of fair clustering algorithms, while suffering only an additional $(1+\varepsilon)$ factor in the approxiation ratio. Setting $\varepsilon = \Omega(1)$ and plugging our coresets into existing algorithms [42, 7, 6], we directly achieve scalable fair clustering algorithms, summarized in Table 2.

We present novel technical ideas to deal with fairness constraints for coresets.

- Our first technical contribution is a reduction to the case $\Gamma = 1$ (Theorem 4.3) which greatly simplifies the problem. Our reduction not only works for our specific construction, but also for all coreset constructions in general.
- Furthermore, to deal with the $\Gamma = 1$ case, we provide several interesting geometric observations for the optimal fair $k$-median/means clustering (Lemma 4.1), which may be of independent interest.

We implement our algorithm and conduct experiments on **Adult**, **Bank**, **Diabetes** and **Athlete** datasets.

- A vanilla implementation results in a coreset with size that depends on $\varepsilon^{-d}$. Our implementation is inspired by our theoretical results and produces coresets whose size is much smaller in practice. This improved implementation is still within the framework of our analysis, and the same worst case theoretical bound still holds.
- To validate the performance of our implementation, we experiment with varying $\varepsilon$ for both fair $k$-median and $k$-means. As expected, the empirical error is well under the theoretical guarantee $\varepsilon$, and the size does not suffer from the $\varepsilon^{-d}$ factor. Specifically, for fair $k$-median, we achieve 5% empirical error using only 3% points of the original data sets, and we achieve similar error using 20% points of the original data set for the $k$-means case. In addition, our coreset for fair $k$-means is better than uniform sampling and that of [42] in the empirical error.

Table 1: Summary of coreset results. $T_1(n)$ and $T_2(n)$ denote the running time of an $O(1)$-approximate algorithm for $k$-median/means, respectively.

| | $k$-Median | | $k$-Means | |
| | size | construction time | size | construction time |
| --- | --- | --- | --- | --- |
| [42] | | | $O(\Gamma k\varepsilon^{-d-2}\log n)$ | $\tilde{O}(k\varepsilon^{-d-2}n\log n + T_2(n))$ |
| This | $O(\Gamma k^2\varepsilon^{-d})$ | $O(k\varepsilon^{-d+1}n + T_1(n))$ | $O(\Gamma k^3\varepsilon^{-d-1})$ | $O(k\varepsilon^{-d+1}n + T_2(n))$ |

Table 2: Summary of fair clustering algorithms. $\Delta$ denotes the maximum number of groups that a point may belong to, and "multi" means the algorithm can handle multiple non-disjoint types.

| | | $k$-Median | | | $k$-Means | |
| | multi | approx. ratio | time | multi | approx. ratio | time |
| --- | --- | --- | --- | --- | --- | --- |
| [20] | | O(1) | $\Omega(n^2)$ | | | |
| [42] | | | | | $O(1)$ | $n^{O(k)}$ |
| [6] | | $\tilde{O}(d\log n)$ | $O(dn\log n + T_1(n))$ | | | |
| [8] | | $(3.488, 1)$ | $\Omega(n^2)$ | | $(4.675, 1)$ | $\Omega(n^2)$ |
| [7] | ✓ | $(O(1), 4\Delta + 4)$ | $\Omega(n^2)$ | ✓ | $(O(1), 4\Delta + 4)$ | $\Omega(n^2)$ |
| This | | $\tilde{O}(d\log n)$ | $O(dlk^2\log(lk) + T_1(lk^2))$ | | $O(1)$ | $(lk)^{O(k)}$ |
| This | ✓ | $(O(1), 4\Delta + 4)$ | $\Omega(l^{2\Delta}k^4)$ | ✓ | $(O(1), 4\Delta + 4)$ | $\Omega(l^{2\Delta}k^6)$ |

- The small size of the coreset translates to more than 200x speed-up (with error ~10%) in the running time of computing the fair clustering objective when the fair constraint $F$ is given. We also apply our coreset on the recent fair clustering algorithm [6, 7], and drastically improve the running time of the algorithm by approximately 5-15 times to [6] and 15-30 times to [7] for all above-mentioned datasets plus a large dataset **Census1990** that consists of 2.5 million records, even taking the coreset construction time into consideration.

## 1.1 Other related works

There are increasingly more works on fair clustering algorithms. Chierichetti et al. [20] introduced the fair clustering problem for a binary type and obtained approximation algorithms for fair $k$-median/center. Backurs et al. [6] improved the running time to nearly linear for fair $k$-median, but the approximation ratio is $\tilde{O}(d\log n)$. Rösner and Schmidt [41] designed a 14-approximate algorithm for fair $k$-center, and the ratio is improved to 5 by [8]. For fair $k$-means, Schmidt et al. [42] introduced the notion of fair coresets, and presented an efficient streaming algorithm. More generally, Bercea et al. [8] proposed a bi-criteria approximation for fair $k$-median/means/center/supplier/facility location. Very recently, Bera et al. [7] presented a bi-criteria approximation algorithm for fair $(k, z)$-clustering problem (Definition 2.3) with arbitrary group structures (potentially overlapping), and Anagnostopoulos et al. [5] improved their results by proposing the first constant-factor approximation algorithm. It is still open to design a near linear time $O(1)$-approximate algorithm for the fair $(k, z)$-clustering problem.

There are other fair variants of clustering problems. Ahmadian et al. [4] studied a variant of the fair $k$-center problem in which the number of each type in each cluster has an upper bound, and proposed a bi-criteria approximation algorithm. Chen et al. [19] studied the fair clustering problem in which any $n/k$ points are entitled to form their own cluster if there is another center closer in distance for all of them. Kleindessner et al. [34] investigate the fair $k$-center problem in which each center has a type, and the selection of the $k$-subset is restricted to include a fixed amount of centers belonging to each type. In another paper [35], they developed fair variants of spectral clusterings (a heuristic $k$-means clustering framework) by incorporating the proportional fairness constraints proposed by [20].

The notion of coreset was first proposed by Agarwal et al. [2]. There has been a large body of work for unconstrained clustering problems in Euclidean spaces [3, 28, 18, 29, 36, 24, 25, 9]). Apart from these, for the general $(k, z)$-clustering problem, Feldman and Langberg [24] presented an $\varepsilon$-coreset of size $\tilde{O}(dk\varepsilon^{-2z})$ in $\tilde{O}(nk)$ time. Huang et al. [30] showed an $\varepsilon$-coreset of size $\tilde{O}(\mathrm{ddim}(X) \cdot k^3\varepsilon^{-2z})$, where $\mathrm{ddim}(X)$ is doubling dimension that measures the intrinsic dimensionality of a space. For

the special case of $k$-means, Braverman et al. [9] improved the size to $\tilde{O}(k\varepsilon^{-2} \cdot \min\{k/\varepsilon, d\})$ by a dimension reduction approach. Works such as [24] use importance sampling technique which avoid the size factor $\varepsilon^{-d}$, but it is unknown if such approaches can be used in fair clustering.

## 2 Problem definition

Consider a set $X \subseteq \mathbb{R}^d$ of $n$ data points, an integer $k$ (number of clusters), and $l$ groups $P_1, \ldots, P_l \subseteq X$. An *assignment constraint*, which was proposed by Schmidt et al. [42], is a $k \times l$ integer matrix $F$. A clustering $\mathcal{C} = \{C_1, \ldots, C_k\}$, which is a $k$-partitioning of $X$, is said to satisfy assignment constraint $F$ if

$$|C_i \cap P_j| = F_{ij}, \ \forall i \in [k], j \in [l].$$

For a $k$-subset $C = \{c_1, \ldots, c_k\} \subseteq X$ (the center set) and $z \in \mathbb{R}_{>0}$, we define $\mathcal{K}_z(X, F, C)$ as the minimum value of $\sum_{i \in [k]} \sum_{x \in C_i} d^z(x, c_i)$ among all clustering $\mathcal{C} = \{C_1, \ldots, C_k\}$ that satisfies $F$, which we call the optimal fair $(k, z)$-clustering value. If there is no clustering satisfying $F$, $\mathcal{K}_z(X, F, C)$ is set to be infinity. The following is our notion of coresets for fair $(k, z)$-clustering. This generalizes the notion introduced in [42] which only considers a partitioned group structure.

**Definition 2.1** (**Coreset for fair clustering**). *Given a set $X \subseteq \mathbb{R}^d$ of $n$ points and $l$ groups $P_1, \ldots, P_l \subseteq X$, a weighted point set $S \subseteq \mathbb{R}^d$ with weight function $w : S \to \mathbb{R}_{>0}$ is an $\varepsilon$-coreset for the fair $(k, z)$-clustering problem, if for each $k$-subset $C \subseteq \mathbb{R}^d$ and each assignment constraint $F \in \mathbb{Z}_{\geq 0}^{k \times l}$, it holds that $\mathcal{K}_z(S, F, C) \in (1 \pm \varepsilon) \cdot \mathcal{K}_z(X, F, C)$.*

Since points in $S$ might receive fractional weights, we change the definition of $\mathcal{K}_z$ a little, so that in evaluating $\mathcal{K}_z(S, F, C)$, a point $x \in S$ may be partially assigned to more than one cluster and the total amount of assignments of $x$ equals $w(x)$.

The currently most general notion of fairness in clustering was proposed by [7], which enforces both upper bounds and lower bounds of any group's proportion in a cluster.

**Definition 2.2** (($\alpha, \beta$)-**proportionally-fair**). *A clustering $\mathcal{C} = (C_1, \ldots, C_k)$ is $(\alpha, \beta)$-proportionally-fair ($\alpha, \beta \in [0, 1]^l$), if for each cluster $C_i$ and $j \in [l]$, it holds that $\alpha_j \leq \frac{|C_i \cap P_j|}{|C_i|} \leq \beta_j$.*

The above definition directly implies for each cluster $C_i$ and any two groups $P_{j_1}, P_{j_2} \in [l]$, $\frac{\alpha_{j_1}}{\beta_{j_2}} \leq \frac{|C_i \cap P_{j_1}|}{|C_i \cap P_{j_2}|} \leq \frac{\beta_{j_1}}{\alpha_{j_2}}$. In other words, the fraction of points belonging to groups $P_{j_1}, P_{j_2}$ in each cluster is bounded from both sides. Indeed, similar fairness constraints have been investigated by works on other fundamental algorithmic problems such as data summarization [14], ranking [16, 44], elections [12], personalization [17, 13], classification [11], and online advertising [15]. Naturally, Bera et al. [7] also defined the fair clustering problem with respect to $(\alpha, \beta)$-proportionally-fairness as follows.

**Definition 2.3** (($\alpha, \beta$)-**proportionally-fair** $(k, z)$-**clustering**). *Given a set $X \subseteq \mathbb{R}^d$ of $n$ points, $l$ groups $P_1, \ldots, P_l \subseteq X$, and two vectors $\alpha, \beta \in [0, 1]^l$, the objective of $(\alpha, \beta)$-proportionally-fair $(k, z)$-clustering is to find a $k$-subset $C = \{c_1, \ldots, c_k\} \in \mathbb{R}^d$ and $(\alpha, \beta)$-proportionally-fair clustering $\mathcal{C} = \{C_1, \ldots, C_k\}$, such that the objective function $\sum_{i \in [k]} \sum_{x \in C_i} d^z(x, c_i)$ is minimized.*

Our notion of coresets is very general, and we relate our notion of coresets to the $(\alpha, \beta)$-proportionally-fair clustering problem, via the following observation, which is similar to Proposition 5 in [42].

**Proposition 2.1.** *Given a $k$-subset $C$, the assignment restriction required by $(\alpha, \beta)$-proportionally-fairness can be modeled as a collection of assignment constraints.*

As a result, if a weighted set $S$ is an $\varepsilon$-coreset satisfying Definition 2.1, then for any $\alpha, \beta \in [0, 1]^l$, the $(\alpha, \beta)$-proportionally-fair $(k, z)$-clustering value computed from $S$ must be a $(1 \pm \varepsilon)$-approximation of that computed from $X$.

## 3 Technical overview

We introduce novel techniques to tackle the assignment constraints. Recall that $\Gamma$ denotes the number of distinct collections of groups that a point may belong to. Our first technical contribution is a general

reduction to the $\Gamma = 1$ case which works for any coreset construction algorithm (Theorem 4.3). The idea is to divide $X$ into $\Gamma$ parts with respect to the groups that a point belongs to, and construct a fair coreset with parameter $\Gamma = 1$ for each group. The observation is that the union of these coresets is a coreset for the original instance and $\Gamma$.

Our coreset construction for the case $\Gamma = 1$ is based on the framework of [29] in which unconstrained $k$-median/means coresets were provided. The main observation of [29] is that it suffices to deal with $X$ that lies on a line. Specifically, they show that it suffices to construct at most $O(k\varepsilon^{-d+1})$ lines, project $X$ to their closest lines and construct an $\varepsilon/3$-coreset for each line. The coreset for each line is then constructed by partitioning the line into $\mathrm{poly}(k/\varepsilon)$ contiguous sub-intervals, and designate at most two points to represent each sub-interval and include these points in the coreset. In their analysis, a crucially used property is that the clustering for any given centers partitions $X$ into $k$ contiguous parts on the line, since each point must be assigned to its nearest center. However, this property might not hold in fair clustering, which is our main difficulty. Nonetheless, we manage to show a new structural lemma, that the optimal fair $k$-median/means clustering partitions $X$ into $O(k)$ contiguous intervals. Specifically, for fair $k$-median, the key geometric observation is that there always exists a center whose corresponding optimal fair $k$-median cluster forms a contiguous interval (Claim 4.1), and this combined with an induction implies the optimal fair clustering partitions $X$ into $2k - 1$ intervals. For fair $k$-means, we show that each optimal fair cluster actually forms a single contiguous interval. Thanks to the new structural properties, plugging in a slightly different set of parameters in [29] yields fair coresets.

## 4 Coresets for fair clustering

For each $x \in X$, denote $\mathcal{P}_x = \{i \in [l] : x \in P_i\}$ as the collection of groups that $x$ belongs to. Let $\Gamma_X$ denote the number of distinct $\mathcal{P}_x$'s, i.e. $\Gamma_X := |\{\mathcal{P}_x : x \in X\}|$. Let $T_z(n)$ denote the running time of a constant approximation algorithm for the $(k, z)$-clustering problem. The main theorems are as follows.

**Theorem 4.1** (**Coreset for fair $k$-median** ($z = 1$)). *There exists an algorithm that constructs an $\varepsilon$-coreset for the fair $k$-median problem of size $O(\Gamma k^2 \varepsilon^{-d})$, in $O(k\varepsilon^{-d+1} n + T_1(n))$ time.*

**Theorem 4.2** (**Coreset for fair $k$-means** ($z = 2$)). *There exists an algorithm that constructs $\varepsilon$-coreset for the fair $k$-means problem of size $O(\Gamma k^3 \varepsilon^{-d-1})$, in $O(k\varepsilon^{-d+1} n + T_2(n))$ time.*

Note that $\Gamma_X$ is usually small. For instance, if there is only one sensitive attribute [42], then each $\mathcal{P}_x$ is singleton and hence $\Gamma_X = l$. More generally, let $\Lambda$ denote the maximum number of groups that any point belongs to, then $\Gamma_X \leq l^\Lambda$, but there is often only $O(1)$ sensitive attributes for each point.

As noted above, the main technical difficulty for the coreset construction is to deal with the assignment constraints. We make an important observation (Theorem 4.3), that one only needs to prove Theorem 4.1 for the case $l = 1$. The proof of Theorem 4.3 can be found in the full version. This theorem is a generalization of Theorem 7 in [42], and the coreset of [42] actually extends to arbitrary group structure thanks to our theorem.

**Theorem 4.3** (**Reduction from $l$ groups to a single group**). *Suppose there exists an algorithm that computes an $\varepsilon$-coreset of size $t$ for the fair $(k, z)$-clustering problem of $\widehat{X}$ with $l = 1$, in time $T(|\widehat{X}|, \varepsilon, k, z)$. Then there exists an algorithm that takes a set $X$, and computes an $\varepsilon$-coreset of size $\Gamma_X \cdot t$ for the fair $(k, z)$-clustering problem, in time $\Gamma_X \cdot T(|X|, \varepsilon, k, z)$.*

Our coreset construction for both fair $k$-median and $k$-means are similar to that in [29], except using a different set of parameters. At a high level, the algorithm reduces general instances to instances where data lie on a line, and it only remains to give a coreset for the line case. Next, we focus on fair $k$-median, and the construction for the $k$-means case is similar and can be found in the full version.

**Remark 4.1.** *Theorem 4.3 can be applied to construct an $\varepsilon$-coreset of size $O(\Gamma_X k\varepsilon^{-d+1})$ for the fair $k$-center clustering problem, since Har-Peled's coreset result [28] directly provides an $\varepsilon$-coreset of size $O(k\varepsilon^{-d+1})$ for the case of $l = 1$.*

### 4.1 The line case

Since $l = 1$, we interpret $F$ as an integer vector in $\mathbb{Z}_{\geq 0}^k$. For a weighted point set $S$ with weight $w : S \to \mathbb{R}_{\geq 0}$, we define the *mean* of $S$ by $\overline{S} := \frac{1}{|S|} \sum_{p \in S} w(p) \cdot p$ and the *error* of $S$ by

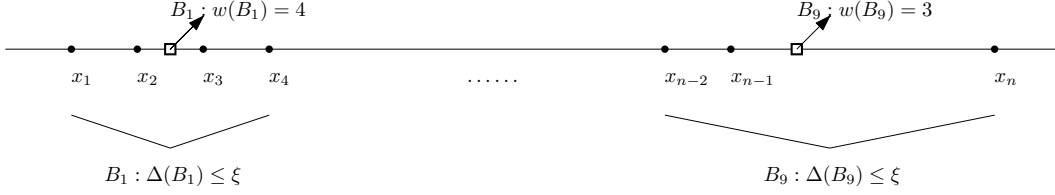

Figure 1: an illustration of Algorithm 1 that divides $X$ into 9 batches.

$\Delta(S) := \sum_{p \in S} w(p) \cdot d(p, \overline{S})$. Denote OPT as the optimal value of the unconstrained $k$-median clustering. Our construction is similar to [29] and we summarize it in Algorithm 1. An illustration of Algorithm 1 may be found in Figure 1.

---

**Input:** $X = \{x_1, \ldots, x_n\} \subset \mathbb{R}^d$ lying on the real line where $x_1 \leq \ldots \leq x_n$, an integer $k \in [n]$, a number OPT as the optimal value of $k$-median clustering.
**Output:** an $\varepsilon$-coreset $S$ of $X$ together with weights $w : S \to \mathbb{R}_{\geq 0}$.
1 Set a threshold $\xi$ satisfying that $\xi = \frac{\varepsilon \cdot \text{OPT}}{30k}$ ;
2 Consider the points from $x_1$ to $x_n$ and group them into batches in a greedy way: each batch $B$ is a maximal point set satisfying that $\Delta(B) \leq \xi$;
3 Denote $\mathcal{B}(X)$ as the collection of all batches. Let $S \leftarrow \bigcup_{B \in \mathcal{B}(X)} \overline{B}$;
4 For each point $x = \overline{B} \in S$, $w(x) \leftarrow |B|$;
5 Return $(S, w)$;

**Algorithm 1:** FairMedian-1D$(X, k)$

---

**Theorem 4.4** (**Coreset for fair $k$-median when $X$ lies on a line**). *Algorithm 1 computes an $\varepsilon/3$-coreset $S$ for fair $k$-median clustering of $X$, in time $O(|X|)$.*

The running time is immediate since for each batch $B \in \mathcal{B}(X)$, it only costs $O(|B|)$ time to compute $\overline{B}$. Hence, Algorithm 1 runs in $O(|X|)$ time. We focus on correctness in the following. In [29], it was shown that $S$ is an $\varepsilon/3$-coreset for the unconstrained $k$-median clustering problem. In their analysis, it is crucially used that the optimal clustering partitions $X$ into $k$ contiguous intervals. Unfortunately, the nice "contiguous" property does not hold in our case because of the assignment constraint $F \in \mathbb{R}^k$. To resolve this issue, we prove a new structural property (Lemma 4.1) that the optimal fair $k$-median clustering actually partitions $X$ into only $O(k)$ contiguous intervals. With this property, Theorem 4.4 is implied by a similar argument as in [29]. The detailed proof can be found in the full version.

**Lemma 4.1** (**Fair $k$-median clustering consists of $2k - 1$ contiguous intervals**). *Suppose $X := \{x_1, \ldots, x_n\} \subset \mathbb{R}^d$ lies on the real line where $x_1 \leq \ldots \leq x_n$. For every $k$-subset $C = (c_1, \ldots, c_k) \in \mathbb{R}^d$ and every assignment constraints $F \in \mathbb{Z}_{\geq 0}^k$, there exists an optimal fair $k$-median clustering that partitions $X$ into at most $2k - 1$ contiguous intervals.*

*Proof.* We prove by induction on $k$. The induction hypothesis is that, for every $k \geq 1$, Lemma 4.1 holds for any data set $X$, any $k$-subset $C \subseteq \mathbb{R}^d$ and any assignment constraint $F \in \mathbb{Z}_{\geq 0}^k$. The base case $k = 1$ holds trivially since all points in $X$ must be assigned to $c_1$.

Assume the lemma holds for $k - 1$ ($k \geq 2$) and we will prove the inductive step $k$. Let $C_1^\star, \ldots, C_k^\star$ be the optimal fair $k$-median clustering w.r.t. $C$ and $F$, where $C_i^\star \subseteq X$ is the subset assigned to center $c_i$. We present the structural property in Claim 4.1, whose proof can be found in the full version.

**Claim 4.1.** *There exists $i \in [k]$ such that $C_i^\star$ consists of exactly one contiguous interval.*

We continue the proof of the inductive step by constructing a reduced instance $(X', F', C')$ where a) $C' := C \setminus \{c_{i_0}\}$; b) $X' = X \setminus C_{i_0}^\star$; c) $F'$ is formed by removing the $i_0$-th coordinate of $F$. Applying the hypothesis on $(X', F', C')$, we know the optimal fair $(k - 1)$-median clustering consists of at

most $2k - 3$ contiguous intervals. Combining with $C_{i_0}^\star$ which has exactly one contiguous interval would increase the number of intervals by at most 2. Thus, we conclude that the optimal fair $k$-median clustering for $(X, F, C)$ has at most $2k - 1$ contiguous intervals. This finishes the inductive step. □

## 4.2 Extending to higher dimension

The extension is the same as that of [29]. We start with a set of $k$ centers that is a $O(1)$-approximate solution $C^\star$ to unconstrained $k$-median. Then we emit $O(\varepsilon^{-d+1})$ rays around each center $c$ in $C^\star$ (which correspond to an $O(\varepsilon)$-net on the unit sphere centered at $c$), and project data points to the nearest ray, such that the total projection cost is $\varepsilon \cdot \mathsf{OPT}/3$. Then for each line, we compute an $\varepsilon/3$-coreset for fair $k$-median by Theorem 4.4, and let $S$ denote the combination of coresets generated from all lines. By the same argument as in Theorem 2.9 of [29], $S$ is an $\varepsilon$-coreset for fair $k$-median clustering, which implies Theorem 4.1. The detailed proof can be found in the full version.

**Remark 4.2.** *In fact, it suffices to emit an arbitrary set of rays such that the total projection cost is at most $\varepsilon \cdot \mathsf{OPT}/3$. This observation is crucially used in our implementations (Section 5) to reduce the size of the coreset, particularly to avoid the construction of the $O(\varepsilon)$-net which is of $O(\varepsilon^{-d})$ size.*

## 5 Empirical results

We implement our algorithm and evaluate its performance on real datasets.[3] The implementation mostly follows our description of algorithms, but a vanilla implementation would bring in an $\varepsilon^{-d}$ factor in the coreset size. To avoid this, as observed in Remark 4.2, we may actually emit any set of rays as long as the total projection cost is bounded, instead of $\varepsilon^{-d}$ rays. We implement this idea by finding the smallest integer $m$ and $m$ lines, such that the minimum cost of projecting data onto $m$ lines is within the error threshold. In our implementation for fair $k$-means, we adopt the widely used Lloyd's heuristic [37] to find the $m$ lines, where the only change to Lloyd's heuristic is that, for each cluster, we need to find a *line* that minimizes the projection cost instead of a point, and we use SVD to efficiently find this line optimally. Unfortunately, the above approach does not work for fair $k$-median, as the SVD does not give the optimal line. As a result, we still need to construct the $\varepsilon$-net, but we alternatively employ some heuristics to find the net adaptively w.r.t. the dataset.

Our evaluation is conducted on four datasets: **Adult** (~50k), **Bank** (~45k) and **Diabetes** (~100k) from UCI Machine Learning Repository [23], and **Athlete** (~200k) from [1], which are also considered in previous papers [20, 42, 7]. For all datasets, we choose numerical features to form a vector in $\mathbb{R}^d$ for each record, where $d = 6$ for **Adult**, $d = 10$ for **Bank**, $d = 29$ for **Diabetes** and $d = 3$ for **Athlete**. We choose two sensitive types for the first three datasets: sex and marital for **Adult** (9 groups, $\Gamma = 14$); marital and default for **Bank** (7 groups, $\Gamma = 12$); sex and age for **Diabetes** (12 groups, $\Gamma = 20$), and we choose a binary sensitive type sex for **Athlete** (2 groups, $\Gamma = 2$). In addition, in the full version, we will also discuss how the following affects the result: a) choosing a binary type as the sensitive type, or b) normalization of the dataset. We pick $k = 3$ (i.e. number of clusters) throughout our experiment. We define the *empirical error* as $\left| \frac{\mathcal{K}_z(S, F, C)}{\mathcal{K}_z(X, F, C)} - 1 \right|$ (which is the same measure as $\varepsilon$) for some $F$ and $C$. To evaluate the empirical error, we draw 500 independent random samples of $(F, C)$ and report the maximum empirical error among these samples. For each $(F, C)$, the fair clustering objectives $\mathcal{K}_z(\cdot, F, C)$ may be formulated as integer linear programs (ILP). We use **CPLEX** [32] to solve the ILP's, report the average running time[4] $T_X$ and $T_S$ for evaluating the objective on dataset $X$ and coreset $S$ respectively, and also report the running time $T_C$ for constructing coreset $S$.

For both $k$-median and $k$-means, we employ *uniform sampling* (**Uni**) as a baseline, in which we partition $X$ into $\Gamma$ parts according to distinct $\mathcal{P}_x$'s (the collection of groups that $x$ belongs to) and take uniform samples from each collection. Additionally, for $k$-means, we select another baseline from a recent work [42] that presented a coreset construction for fair $k$-means, whose implementation is based on the **BICO** library which is a high-performance coreset-based library for computing k-means clustering [26]. We evaluate the performance of our coreset for fair $k$-means against **BICO** and **Uni**. As a remark of **BICO** and **Uni** implementations, they do not support specifying parameter $\varepsilon$, but a hinted size of the resulted coreset. Hence, we start with evaluating our coreset, and set the hinted size for **Uni** and **BICO** as the size of our coreset.

We also showcase the speed-up to two recently published approximation algorithms by applying a 0.5-coreset. The first algorithm is a practically efficient, $O(\log n)$-approximate algorithm for fair $k$-median [6] that works for a binary type, referred to as **FairTree**. The other one is a bi-criteria approximation algorithm [7] for both fair $k$-median and $k$-means, referred to as **FairLP**. We slightly modify the implementations of **FairTree** and **FairLP** to enable them work with our coreset, particularly making them handle weighted inputs efficiently. We do experiments on a large dataset **Census1990** which consists of about 2.5 million records (where we select $d = 13$ features and a binary type), in addition to the above-mentioned **Adult**, **Bank**, **Diabetes** and **Athlete** datasets.

Table 3: performance of $\varepsilon$-coresets for fair $k$-median w.r.t. varying $\varepsilon$.

|  | $\varepsilon$ | emp. err. Ours | emp. err. **Uni** | size | $T_S$ (ms) | $T_C$ (ms) | $T_X$ (ms) |
|---|---|---|---|---|---|---|---|
| **Adult** | 10% | 2.36% | 12.28% | 262 | 13 | 408 | 7101 |
|  | 20% | 4.36% | 17.17% | 215 | 12 | 311 | - |
|  | 30% | 4.46% | 15.12% | 161 | 9 | 295 | - |
|  | 40% | 8.52% | 31.96% | 139 | 9 | 282 | - |
| **Bank** | 10% | 1.45% | 5.32% | 2393 | 111 | 971 | 5453 |
|  | 20% | 2.24% | 3.38% | 1101 | 50 | 689 | - |
|  | 30% | 4.18% | 14.60% | 506 | 24 | 476 | - |
|  | 40% | 5.35% | 10.53% | 293 | 14 | 452 | - |
| **Diabetes** | 10% | 0.55% | 6.38% | 85822 | 12112 | 141212 | 17532 |
|  | 20% | 1.62% | 15.44% | 34271 | 3267 | 16040 | - |
|  | 30% | 3.61% | 1.92% | 6693 | 411 | 5017 | - |
|  | 40% | 5.33% | 3.67% | 2949 | 160 | 3916 | - |
| **Athlete** | 10% | 1.14% | 2.87% | 3959 | 96 | 8141 | 74851 |
|  | 20% | 2.59% | 4.38% | 685 | 19 | 3779 | - |
|  | 30% | 4.86% | 4.98% | 316 | 11 | 2763 | - |
|  | 40% | 8.25% | 16.59% | 112 | 7 | 2390 | - |

Table 4: performance of $\varepsilon$-coresets for fair $k$-means w.r.t. varying $\varepsilon$.

|  | $\varepsilon$ | emp. err. Ours | emp. err. **BICO** | emp. err. **Uni** | size | $T_S$ (ms) | $T_C$ (ms) Ours | $T_C$ (ms) **BICO** | $T_X$ (ms) |
|---|---|---|---|---|---|---|---|---|---|
| **Adult** | 10% | 0.28% | 1.04% | 10.63% | 880 | 44 | 1351 | 786 | 7404 |
|  | 20% | 0.55% | 1.12% | 2.87% | 610 | 29 | 511 | 788 | - |
|  | 30% | 1.17% | 4.06% | 19.91% | 503 | 26 | 495 | 750 | - |
|  | 40% | 2.20% | 4.45% | 48.10% | 433 | 22 | 492 | 768 | - |
| **Bank** | 10% | 2.85% | 2.71% | 30.68% | 409 | 19 | 507 | 718 | 5128 |
|  | 20% | 2.93% | 4.59% | 45.09% | 280 | 14 | 478 | 712 | - |
|  | 30% | 2.68% | 6.10% | 24.82% | 230 | 11 | 531 | 711 | - |
|  | 40% | 2.30% | 5.66% | 33.42% | 194 | 10 | 505 | 690 | - |
| **Diabetes** | 10% | 4.39% | 10.54% | 1.91% | 50163 | 5300 | 65189 | 2615 | 16312 |
|  | 20% | 11.24% | 11.32% | 4.41% | 3385 | 168 | 5138 | 1544 | - |
|  | 30% | 14.52% | 20.54% | 13.46% | 958 | 44 | 2680 | 1480 | - |
|  | 40% | 13.95% | 22.05% | 10.92% | 775 | 35 | 2657 | 1462 | - |
| **Athlete** | 10% | 5.43% | 4.94% | 10.96% | 1516 | 36 | 14534 | 1160 | 73743 |
|  | 20% | 11.41% | 21.31% | 10.62% | 213 | 9 | 3566 | 1090 | - |
|  | 30% | 13.18% | 29.97% | 16.93% | 98 | 7 | 2591 | 1076 | - |
|  | 40% | 13.01% | 29.74% | 152.31% | 83 | 6 | 2613 | 1066 | - |

## 5.1 Results

Table 3 and 4 summarize the accuracy-size trade-off of our coresets for fair $k$-median and $k$-means respectively, under different error guarantee $\varepsilon$. Since the coreset construction time $T_C$ for **Uni** is very small (usually less than 50 ms) we do not report it in the table. From the table, a key finding is that the size of the coreset does not suffer from the $\varepsilon^{-d}$ factor thanks to our optimized implementation.

Table 5: speed-up of fair clustering algorithms using our coreset. $\text{obj}_{\text{ALG}}/\text{obj}_{\text{ALG}}$ is the runtime/clustering objective w/o our coreset and $T'_{\text{ALG}}/\text{obj}'_{\text{ALG}}$ is the runtime/clustering objective on top of our coreset.

| | ALG | $\text{obj}_{\text{ALG}}$ | $\text{obj}'_{\text{ALG}}$ | $T_{\text{ALG}}$ (s) | $T'_{\text{ALG}}$ (s) | $T_{\text{C}}$ (s) |
|---|---|---|---|---|---|---|
| **Adult** | **FairTree** ($z=1$) | $2.09 \times 10^9$ | $1.23 \times 10^9$ | 12.62 | 0.38 | 0.63 |
| | **FairLP** ($z=2$) | $1.23 \times 10^{14}$ | $1.44 \times 10^{14}$ | 19.92 | 0.20 | 1.03 |
| **Bank** | **FairTree** ($z=1$) | $5.69 \times 10^6$ | $4.70 \times 10^6$ | 14.62 | 0.64 | 0.60 |
| | **FairLP** ($z=2$) | $1.53 \times 10^9$ | $1.46 \times 10^9$ | 17.41 | 0.08 | 0.50 |
| **Diabetes** | **FairTree** ($z=1$) | $1.13 \times 10^6$ | $9.50 \times 10^5$ | 19.26 | 1.70 | 2.96 |
| | **FairLP** ($z=2$) | $1.47 \times 10^7$ | $1.08 \times 10^7$ | 55.11 | 0.41 | 2.61 |
| **Athlete** | **FairTree** ($z=1$) | $2.50 \times 10^6$ | $2.42 \times 10^6$ | 29.94 | 1.34 | 2.35 |
| | **FairLP** ($z=2$) | $3.33 \times 10^7$ | $2.89 \times 10^7$ | 37.50 | 0.03 | 2.42 |
| **Census1990** | **FairTree** ($z=1$) | $9.38 \times 10^6$ | $7.65 \times 10^6$ | 450.79 | 23.36 | 20.28 |
| | **FairLP** ($z=2$) | $4.19 \times 10^7$ | $1.32 \times 10^7$ | 1048.72 | 0.06 | 31.05 |

As for the fair $k$-median, the empirical error of our coreset is well under control. In particular, to achieve 5% empirical error, only less than 3 percents of data is necessary for all datasets, and this results in a ~200x acceleration in evaluating the objective and 10x acceleration even taking the coreset construction time into consideration.[5] Regarding the running time, our coreset construction time scales roughly linearly with the size of the coreset, which means our algorithm is output-sensitive. The empirical error of **Uni** is comparable to ours on **Diabetes**, but the worst-case error is unbounded (2x-10x to our coreset, even larger than $\varepsilon$) in general and seems not stable when $\varepsilon$ varies.

Our coreset works well for fair $k$-means, and it also offers significant acceleration of evaluating the objective. Compared with **BICO**, our coreset achieves smaller empirical error for fixed $\varepsilon$ and the construction time is between 0.5x to 2x that of **BICO**. Again, the empirical error of **Uni** could be 2x smaller than ours and **BICO** on **Diabetes**, but the worst-case error is unbounded in general.

Table 5 demonstrates the speed-up to **FairTree** and **FairLP** with the help of our coreset. We observed that the adaption of our coresets offers a 5x-15x speed-up to **FairTree** and a 15x-30x speed-up to **FairLP** for all datasets, even taking the coreset construction time into consideration. Specifically, the runtime on top of our coreset for **FairLP** is less than 1s for all datasets, which is extremely fast. We also observe that the clustering objective $\text{obj}'_{\text{ALG}}$ on top of our coresets is usually within 0.6-1.2 times of $\text{obj}_{\text{ALG}}$ which is the objective without the coreset (noting that coresets might shrink the objective). The only exception is **FairLP** on **Census1990**, in which $\text{obj}'_{\text{ALG}}$ is only 35% of $\text{obj}_{\text{ALG}}$. A possible reason is that in the implementation of **FairLP**, an important step is to compute an approximate (unconstrained) $k$-means clustering solution on the dataset by employing the *sklearn* library [39]. However, *sklearn* tends to trade accuracy for speed when the dataset gets large. As a result, **FairLP** actually finds a better approximate $k$-means solution on the coreset than on the large dataset **Census1990** and hence applying coresets can achieve a much smaller clustering objective.

## 6 Future work

This paper constructs $\varepsilon$-coresets for the fair $k$-median/means clustering problem of size independent on the full dataset, and when the data may have multiple, non-disjoint types. Our coreset for fair $k$-median is the first known coreset construction to the best of our knowledge. For fair $k$-means, we improve the coreset size of the prior result [42], and extend it to multiple non-disjoint types. The empirical results show that our coresets are indeed much smaller than the full dataset and result in significant reductions in the running time of computing the fair clustering objective.

Our work leaves several interesting futural directions. For unconstrained clustering, there exist several works using the sampling approach such that the coreset size does not depend exponentially on the Euclidean dimension $d$. It is interesting to investigate whether sampling approaches can be applied for constructing fair coresets and achieve similar size bound as the unconstrained setting. Another direction is to construct coresets for general fair $(k, z)$-clustering beyond $k$-median/means/center.

## Acknowledgments

This research was supported in part by NSF CCF-1908347, SNSF 200021_182527, ONR Award N00014-18-1-2364 and a Minerva Foundation grant.

## Footnotes

[3] `https://github.com/sfjiang1990/Coresets-for-Clustering-with-Fairness-Constraints`.

[4] The experiments are conducted on a 4-Core desktop CPU with 64 GB RAM.

[5]The same coreset may be used for clustering with any assignment constraints, so its construction time would be averaged out if multiple fair clustering tasks are performed.

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
