[Supplementary Material]



# A  Missing details in Section 4

## A.1  Proof of Theorem 4.2

*Proof.* Consider the case that $\Gamma = 1$ in which all $\mathcal{P}_x$s are the same. Hence, this case can be degenerated to $l = 1$ and has an $\varepsilon$-coreset of size $t$ by assumption. Divide the point set $X$ into $X^{(1)}, \ldots, X^{(\Gamma)}$ by $\mathcal{P}_x$, i.e., for each $i \in [\Gamma]$, all collections $\mathcal{P}_x$ ($x \in X^{(i)}$) are the same, denoted by $\mathcal{P}_i$. For each $i \in [\Gamma]$, suppose $S^{(i)}$ is an $\varepsilon$-coreset for the fair $(k, z)$-clustering problem of $X^{(i)}$ where each point in $S^{(i)}$ belongs to all groups in $\mathcal{P}_i$. Let $S := \bigcup_{i \in [l]} S^{(i)}$. It is sufficient to prove $S$ is an $\varepsilon$-coreset for the fair $(k, z)$-clustering problem of $X$.

Given a $k$-subset $C \subseteq \mathbb{R}^d$ and an assignment constraint $F$, let $C_1^\star, \ldots, C_k^\star$ be the optimal fair clustering of the instance $(X, F, C)$. Then for each collection $X^{(i)}$ ($i \in [\Gamma]$), we construct an assignment constraint $F^{(i)} \in \mathcal{Z}^{k \times l}$ as follows: for each $j_1 \in [k]$ and $j_2 \in [l]$, let $F_{j_1, j_2}^{(i)} = 0$ if $j_2 \notin \mathcal{P}_i$ and $\left|C_{j_1}^\star \cap X^{(i)}\right|$ if $j_2 \in \mathcal{P}_i$, i.e., $F_{j_1, j_2}^{(i)}$ is the number of points within $X^{(i)}$ that belong to $C_{j_1} \cap P_{j_2}$. By definition, we have that for each $j_1 \in [k]$ and $j_2 \in [l]$,

$$F_{j_1, j_2} = \sum_{i \in [\Gamma]} F_{j_1, j_2}^{(i)}. \tag{1}$$

Then

$$\mathcal{K}_z(X, F, C) = \sum_{i \in [l]} \mathcal{K}_z(X^{(i)}, F^{(i)}, C) \qquad \text{(Defns. of } \mathcal{K}_z \text{ and } F^{(i)})$$

$$\geq (1 - \varepsilon) \cdot \sum_{i \in [l]} \mathcal{K}_z(S^{(i)}, F^{(i)}, C) \qquad \text{(Defn. of } S^{(i)})$$

$$\geq (1 - \varepsilon) \cdot \mathcal{K}_z(S, F, C) \qquad \text{(Optimality and Eq. (1)).}$$

Similarly, we can prove that $\mathcal{K}_z(S, F, C) \geq (1 - \varepsilon)\mathcal{K}_z(X, F, C)$. It completes the proof. $\qquad \square$

## A.2  Proof of Claim 4.1

*Proof.* We first prove the following fact for preparation.

**Fact A.1.** *Suppose $p, q \in \mathbb{R}^d$. Define $f : \mathbb{R} \to \mathbb{R}$ as $f(x) := d(x, p) - d(x, q)$ (here we abuse the notation by treating $x$ as a point in the $x$-axis of $\mathbb{R}^d$). Then $f$ is either ID or DI.*[3]

*Proof.* Let $h_p$ and $h_q$ be the distance from $p$ and $q$ to the x-axis respectively, and let $u_p$ and $u_q$ be the corresponding $x$-coordinate of $p$ and $q$. We have

$$f(x) = \sqrt{(x - u_p)^2 + h_p^2} - \sqrt{(x - u_q)^2 + h_q^2}.$$

Then we can regard $p, q$ as two points in $\mathbb{R}^2$ by letting $p = (u_p, h_p)$ and $q = (u_q, h_q)$. Also we have

$$f'(x) = \frac{x - u_p}{\sqrt{(x - u_p)^2 + h_p^2}} - \frac{x - u_q}{\sqrt{(x - u_q)^2 + h_q^2}} = \frac{x - u_p}{d(x, p)} - \frac{x - u_q}{d(x, q)}.$$

W.l.o.g. assume that $u_p \leq u_q$. Next, we rewrite $f'(x)$ with respect to $\cos(\angle pxu_p)$ and $\cos(\angle qxu_q)$.

1. If $x \leq u_p$. Then $f'(x) = \frac{d(x, u_q)}{d(x, q)} - \frac{d(x, u_p)}{d(x, p)} = \cos(\angle qxu_q) - \cos(\angle pxu_p)$.

2. If $u_p < x \leq u_q$. Then $f'(x) = \frac{d(x, u_p)}{d(x, p)} + \frac{d(x, u_q)}{d(x, q)} = \cos(\angle pxu_p) + \cos(\angle qxu_q)$.

3. If $x > u_q$. Then $f'(x) = \frac{d(x, u_p)}{d(x, p)} - \frac{d(x, u_q)}{d(x, q)} = \cos(\angle pxu_p) - \cos(\angle qxu_q)$.

Denote the intersecting point of line $pq$ and the $x$-axis to be $y$. Specifically, if $h_p = h_q$, we denote $y = -\infty$. Note that $f'(x) = 0$ if and only if $x = y$. Now we analyze $f'(x)$ in two cases (whether or not $h_p \leq h_q$).

- Case i): $h_p \leq h_q$ which implies that $y < u_p$. When $x$ goes from $-\infty$ to $u_p$, first $f'(x) \leq 0$ and then $f'(x) \geq 0$. When $x > u_p$, $f'(x) \geq 0$.

- Case ii): $h_p > h_q$ which implies that $y > u_q$. When $x \leq u_q$, $f'(x) \geq 0$. When $x$ goes from $u_q$ to $+\infty$, first $f'(x) \geq 0$ and then $f'(x) \leq 0$.

Therefore, $f(x)$ is either DI or ID. $\qquad\square$

Suppose for the contrary that for any $i \in [k]$, $C_i^\star$ consists of at least two contiguous intervals. Pick any $i$ and suppose $S_L, S_R \subseteq C_i^\star$ are two contiguous intervals such that $S_L$ lies on the left of $S_R$. Let $y_L$ denote the rightmost point of $S_L$ and $y_R$ denote the leftmost point of $S_R$. Since $S_L$ and $S_R$ are two distinct contiguous intervals, there exists some point $y \in X$ between $y_L$ and $y_R$ such that $y \in C_j^\star$ for some $j \neq i$. Define $g : \mathbb{R} \to \mathbb{R}$ as $g(x) := d(x, c_j) - d(x, c_i)$. By Fact A.1, we know that $g(x)$ is either ID or DI.

If $g$ is ID, we swap the assignment of $y$ and $y_{\min} := \arg\min_{x \in \{y_L, y_R\}} g(x)$ in the optimal fair $k$-median clustering. Since $g$ is ID, for any interval $P$ with endpoints $p$ and $q$, $\min_{x \in P} g(x) = \min_{x \in \{p,q\}} g(x)$. This fact together with $y_L \leq y \leq y_R$ implies that $g(y_{\min}) - g(y) \leq 0$. Hence, the change of the objective is

$$d(y, c_i) - d(y, c_j) - d(y_{\min}, c_i) + d(y_{\min}, c_j) = g(y_{\min}) - g(y) \leq 0.$$

This contradicts with the optimality of $C^\star$ and hence $g$ has to be DI.

Next, we show that there is no $y' \in C_j^\star$ such that $y' < y_L$ or $y' > y_R$. We prove by contradiction and only focus on the case of $y' < y_L$, since the case of $z > y_R$ can be proved similarly by symmetry. We swap the assignment of $y_L$ and $y_{\max} := \arg\max_{x \in \{y, y'\}} g(x)$ in the optimal fair $k$-median clustering. The change of the objective is

$$d(y_L, c_j) - d(y_L, c_i) - d(y_{\max}, c_j) + d(y_{\max}, c_i)$$
$$= g(y_L) - g(y_{\max}) \leq 0,$$

where the last inequality is by the fact that $g$ is DI. This contradicts the optimality of $C^\star$. Hence, we conclude such $y'$ does not exist.

Therefore, $\forall x \in C_j^\star, y_L < x < y_R$. By assumption, $C_j^\star$ consists of at least two contiguous intervals within $(y_L, y_R)$. However, we can actually do exactly the same argument for $C_j^\star$ as in the $i$ case, and eventually we would find a $j'$ such that $C_{j'}^\star$ lies inside a strict smaller interval $(y_L', y_R')$ of $X$, where $y_L < y_L' < y_R' < y_R$. Since $n$ is finite, we cannot do this procedure infinitely, which is a contradiction. This finishes the proof of Claim 4.1. $\qquad\square$

## A.3 Details of Section 4.2

For completeness, we describe the detailed procedure for coresets for fair $k$-median.

1. We start with computing an approximate $k$-subset $C^\star = \{c_1, \ldots, c_k\} \subseteq \mathbb{R}^d$ such that $\mathsf{OPT} \leq \mathcal{K}_2(X, C^\star) \leq c \cdot \mathsf{OPT}$ for some constant $c > 1$.[4]

2. Then we partition the point set $X$ into sets $X_1, \ldots, X_k$ satisfying that $X_i$ is the collection of points closest to $c_i$.

3. For each center $c_i$, we take a unit sphere centered at $c_i$ and construct an $\frac{\varepsilon}{3c}$-net $N_{c_i}$[5] on this sphere. By Lemma 2.6 in [25], $|N_{c_i}| = O(\varepsilon^{-d+1})$ and may be computed in $O(\varepsilon^{-d+1})$ time. Then for every $p \in N_{c_i}$, we emit a ray from $c_i$ to $p$. Overall, there are at most $O(k\varepsilon^{-d+1})$ lines.

4. For each $i \in [k]$, we project all points of $X_i$ onto the closest line around $c_i$. Let $\pi : X \rightarrow \mathbb{R}^d$ denote the projection function. By the definition of $\frac{\varepsilon}{3c}$-net, we have that $\sum_{x \in X} d(x, \pi(x)) \leq \varepsilon \cdot \text{OPT}/3$ which indicates that the projection cost is negligible. Then for each line, we compute an $\varepsilon/3$-coreset of size $O(k\varepsilon^{-1})$ for fair $k$-median by Theorem 4.3. Let $S$ denote the combination of coresets generated from all lines.

# B  Full version of Section 5

In this section, we provide the details of coreset construction for fair $k$-means clustering. Recall that the main theorem is as follows.

**Theorem B.1** (**Coreset for fair $k$-means**). *There exists an algorithm that constructs $\varepsilon$-coreset for the fair $k$-means problem of size $O(\Gamma k^3 \varepsilon^{-d-1})$, in $O(k^2 \varepsilon^{-d+1} n + T_2(n, d, k))$ time.*

Note that the above result improves the coreset size of [36] by a $O(\frac{\log n}{\varepsilon k^2})$ factor. Similar to the fair $k$-median case, it suffices to prove for the case $l = 1$. Recall that an assignment constraint for $l = 1$ can be described by a vector $F \in \mathbb{R}^k$. Denote OPT to be the optimal $k$-means value without any assignment constraint.

## B.1  The line case

Similar to [25], we first consider the case that $X$ is a point set on the real line. For a weighted point set $S$ with weight $w : S \rightarrow \mathbb{R}_{\geq 0}$, we denote the *mean* of $S$ by $\overline{S} := \frac{1}{|S|} \sum_{p \in S} w(p) \cdot p$, and the *error* of $S$ by $\Delta(S) := \sum_{p \in S} w(p) \cdot d^2(p, \overline{S})$.

**Construction.**    Same to [25], we consider the points from left to right and group them into batches in a greedy way: each batch $B$ is a maximal point set satisfying that $\Delta(B) \leq \xi$ where $\xi = \frac{\varepsilon^2 \text{OPT}}{200k^2}$. Let $\mathcal{I}(B)$ denote the smallest closed segment containing all the points of a batch $B$. Let $\mathcal{B}(X)$ denote the collection of all batches. For each batch $B$, we construct a collection $\mathcal{J}(B)$ of two weighted points satisfying Lemma 5.1. The coreset is defined by $S = \bigcup_{B \in \mathcal{B}(X)} \mathcal{J}(B)$.

**Lemma B.1** (**Lemmas 3.2 and 3.4 in [25]**). *The number of batches is $O(k^2/\varepsilon^2)$. For each batch $B$, there exist two weighted points $q_1, q_2 \in \mathcal{I}(B)$ together with weight $w_1, w_2$ satisfying that*

- $w_1 + w_2 = |B|$.

- *Let $\mathcal{J}(B)$ denote the collection of two weighted points $q_1$ and $q_2$. Then we have $\overline{\mathcal{J}(B)} = \overline{B}$ and $\Delta(B) = \Delta(\mathcal{J}(B))$.*

- *Given any point $q \in \mathbb{R}^d$, we have*
$$\mathcal{K}_2(B, q) = \Delta(B) + |B| \cdot d^2(q, \overline{B}) = \mathcal{K}_2(\mathcal{J}(B), q).$$

**Analysis.**    We argue that $S$ is indeed an $\varepsilon/3$-coreset for the fair $k$-means clustering problem. By Theorem 3.5 in [25], $S$ is an $\varepsilon/3$-coreset for $k$-means clustering of $X$. However, we need to handle additional assignment constraints. To address this, we introduce the following lemma showing that every optimal cluster satisfying the given assignment constraint is within a contiguous interval.

**Lemma B.2** (**Clusters are contiguous for fair $k$-means**). *Suppose $X = \{x_1, \ldots, x_n\}$ where $x_1 \leq x_2 \leq \ldots \leq x_n$. Given an assignment constraint $F \in \mathbb{R}^k$ and a $k$-subset $C = \{c_1, \ldots, c_k\} \subseteq \mathbb{R}^d$. Then letting $C_i := \left\{ x_{1 + \sum_{j < i} F_j}, \ldots, x_{\sum_{j \leq i} F_j} \right\}$ ($i \in [k]$), we have*
$$\mathcal{K}_2(X, F, C) = \sum_{i \in [k]} \sum_{x \in C_i} d^2(x, c_i).$$

*Proof.* Let $c_i'$ denote the projection of point $c_i$ to the real line and assume that $c_1' \leq c_2' \leq \ldots \leq c_k'$. We slightly abuse the notation by regarding point $c_i'$ as a real value. We prove the lemma by contradiction. Let $C_1, \ldots, C_k$ be the optimal fair clustering. By contradiction we assume that there exists $i_1 < i_2$ and $j_1 < j_2$ such that $x_{j_1} \in C_{i_2}$ and $x_{j_2} \in C_{i_1}$. By the definitions of $c_{i_1}'$ and $c_{i_2}'$, we have that
$$d(c_{i_1}', x_{j_1}) + d(c_{i_2}', x_{j_2}) \leq d(c_{i_1}', x_{j_2}) + d(c_{i_2}', x_{j_1}), \tag{2}$$

and

$$\max\left\{d(c'_{i_1}, x_{j_1}), d(c'_{i_2}, x_{j_2})\right\} \leq \max\left\{d(c'_{i_1}, x_{j_2}), d(c'_{i_2}, x_{j_1})\right\}. \tag{3}$$

Combining Inequalities (2) and (3), we argue that

$$d^2(c'_{i_1}, x_{j_1}) + d^2(c'_{i_2}, x_{j_2}) \leq d^2(c'_{i_1}, x_{j_2}) + d^2(c'_{i_2}, x_{j_1}) \tag{4}$$

by proving the following claim.

**Claim B.1.** *Suppose $a, b, c, d \geq 0$, $a + b \leq c + d$ and $a, b, c \leq d$. Then $a^2 + b^2 \leq c^2 + d^2$.*

*Proof.* If $a + b \leq d$, then we have $a^2 + b^2 \leq (a+b)^2 \leq d^2 \leq c^2 + d^2$. So we assume that $a + b > d$. Let $e = a + b - d > 0$. Since $a + b \leq c + d$, we have $e^2 \leq c^2$. Hence, it suffices to prove that $a^2 + b^2 \leq e^2 + d^2$. Note that

$$e^2 + d^2 = (a + b - d)^2 + d^2 = a^2 + b^2 + (d - a)(d - b) \geq a^2 + b^2,$$

which completes the proof. $\qquad\square$

Now we come back to prove Lemma B.1. We have the following inequality.

$$
\begin{aligned}
& d^2(x_{j_1}, c_{i_1}) + d^2(x_{j_2}, c_{i_2}) \\
=& d^2(x_{j_1}, c'_{i_1}) + d^2(c'_{i_1}, c_{i_1}) + d^2(x_{j_2}, c'_{i_2}) + d^2(c'_{i_2}, c_{i_2}) && \text{(The Pythagorean theorem)} \\
\leq& d^2(x_{j_1}, c'_{i_2}) + d^2(c'_{i_1}, c_{i_1}) + d^2(x_{j_2}, c'_{i_1}) + d^2(c'_{i_2}, c_{i_2}) && \text{(Ineq. (4))} \\
=& d^2(x_{j_1}, c_{i_2}) + d^2(x_{j_2}, c_{i_1}). && \text{(The Pythagorean theorem)}
\end{aligned}
$$

It contradicts with the assumption that $x_{j_1} \in C_{i_2}$ and $x_{j_2} \in C_{i_1}$. Hence, we complete the proof. $\quad\square$

Now we are ready to give the following theorem.

**Theorem B.2 (Coreset for fair $k$-means when $X$ lies on a line).** *Let $X$ be a set of $n$ points lying on a line in $\mathbb{R}^d$. Let $S = \bigcup_{B \in \mathcal{B}(X)} \mathcal{J}(B)$ be the coreset constructed as in Lemma B.1. Then $S$ is an $\varepsilon/3$-coreset for fair $k$-means clustering of $X$.*

*Proof.* The proof is similar to that of Theorem 3.5 in [25]. The running time analysis is exactly the same. Hence, we only focus on the correctness analysis in the following. We first rotate space such that the line is on the $x$-axis and assume that $x_1 \leq x_2 \leq \ldots \leq x_n$. Given an assignment constraint $F \in \mathbb{R}^k$ and a $k$-subset $C = \{c_1, \ldots, c_k\} \subseteq \mathbb{R}^d$, let $c'_i$ denote the projection of point $c_i$ to the real line and assume that $c'_1 \leq c'_2 \leq \ldots \leq c'_k$. Our goal is to prove that

$$|\mathcal{K}_2(S, F, C) - \mathcal{K}_2(X, F, C)| \leq \frac{\varepsilon}{3} \cdot \mathcal{K}_2(X, F, C).$$

By Lemma B.2, we have that the optimal fair clustering of $X$ should be $C_i := \left\{x_{1+\sum_{j<i} F_j}, \ldots, x_{\sum_{j \leq i} F_j}\right\}$ for each $i \in [k]$. Hence, $\mathcal{I}(C_1), \ldots, \mathcal{I}(C_k)$ are disjoint intervals. Similarly, the optimal fair clustering of $X$ should be to scan weighted points in $S$ from left to right and cluster points of total weight $F_i$ to $c_i$.[6] If a batch $B \in \mathcal{B}(X)$ lies completely within some interval $\mathcal{I}(C_i)$, then it does not contribute to the overall difference $|\mathcal{K}_2(S, F, C) - \mathcal{K}_2(X, F, C)|$ by Lemma B.1.

Thus, the only problematic batches are those that contain an endpoint of $\mathcal{I}(C_1), \ldots, \mathcal{I}(C_k)$. There are at most $k - 1$ such batches. Let $B$ be one such batch and $\mathcal{J}(B) = \{q_1, q_2\}$ be constructed as in Lemma B.1. For $i \in [k]$, let $V_i := \mathcal{I}(C_i) \cap B$. Let $T$ denote the collection of the $w_1$ left side points within $B$ and $T' = B \setminus T$. Note that $w_1$ may be fractional and hence $T$ may include a fractional point. Denote

$$\eta := \sum_{i \in [k]} \sum_{x \in V_i \cap T} d^2(x, q_1) + \sum_{i \in [k]} \sum_{x \in V_i \cap T'} d^2(x, q_2).$$

We have that

$$
\begin{aligned}
\eta &= \sum_{i\in[k]}\sum_{x\in V_i\cap T}\left(d(x,\overline{B})-d(q_1,\overline{B})\right)^2 + \sum_{i\in[k]}\sum_{x\in V_i\cap T'}\left(d(x,\overline{B})-d(q_2,\overline{B})\right)^2 \\
&\leq \sum_{i\in[k]}\sum_{x\in V_i\cap T}\left(d^2(x,\overline{B})+d^2(q_1,\overline{B})\right) + \sum_{i\in[k]}\sum_{x\in V_i\cap T'}\left(d^2(x,\overline{B})+d^2(q_2,\overline{B})\right) \\
&=\Delta(B)+\Delta(\mathcal{J}(B))=2\Delta(B) \qquad\qquad \text{(Lemma B.1)} \\
&\leq \frac{\varepsilon^2\mathsf{OPT}}{100k} \qquad\qquad \text{(Construction of } B\text{).}
\end{aligned}
\tag{5}
$$

Then we can upper bound the contribution of $B$ to the overall difference $|\mathcal{K}_2(S,F,C)-\mathcal{K}_2(X,F,C)|$ by

$$
\begin{aligned}
&\left|\sum_{i\in[k]}\sum_{x\in V_i\cap T}\left(d^2(x,c_i)-d^2(q_1,c_i)\right) + \sum_{i\in[k]}\sum_{x\in V_i\cap T'}\left(d^2(x,c_i)-d^2(q_2,c_i)\right)\right| \\
&\leq \sum_{i\in[k]}\sum_{x\in V_i\cap T}\left|d^2(x,c_i)-d^2(q_1,c_i)\right| + \sum_{i\in[k]}\sum_{x\in V_i\cap T'}\left|d^2(x,c_i)-d^2(q_2,c_i)\right| \\
&= \sum_{i\in[k]}\sum_{x\in V_i\cap T}d(x,q_1)\left(d(x,c_i)+d(q_1,c_i)\right) + \sum_{i\in[k]}\sum_{x\in V_i\cap T'}d(x,q_2)\left(d(x,c_i)+d(q_2,c_i)\right) \\
&\leq \sum_{i\in[k]}\sum_{x\in V_i\cap T}d(x,q_1)\left(2d(x,c_i)+d(x,q_1)\right) + \sum_{i\in[k]}\sum_{x\in V_i\cap T'}d(x,q_2)\left(2d(x,c_i)+d(x,q_2)\right) \\
&= \sum_{i\in[k]}\sum_{x\in V_i\cap T}d^2(x,q_1) + \sum_{i\in[k]}\sum_{x\in V_i\cap T'}d^2(x,q_2) \\
&\quad + 2\sum_{i\in[k]}\sum_{x\in V_i\cap T}d(x,q_1)d(x,c_i) + 2\sum_{i\in[k]}\sum_{x\in V_i\cap T'}d(x,q_2)d(x,c_i) \\
&\leq \eta + 2\sqrt{\eta}\sqrt{\sum_{i\in[k]}\sum_{x\in V_i}d^2(x,c_i)} \qquad\qquad \text{(Defn. of } \eta \text{ and Cauchy-Schwarz)} \\
&\leq \frac{\varepsilon^2\mathsf{OPT}}{50k}+\frac{2\varepsilon}{7k}\sqrt{\mathsf{OPT}\cdot\mathcal{K}_2(X,F,C)} \qquad\qquad \text{(Ineq. (5))} \\
&\leq \frac{\varepsilon^2\mathsf{OPT}}{100k}+\frac{2\varepsilon}{10k}\cdot\frac{\mathsf{OPT}+\sum_{i\in[k]}\sum_{x\in V_i}d^2(x,c_i)}{2} \\
&\leq \frac{\varepsilon\mathsf{OPT}}{5k}+\frac{\varepsilon\sum_{i\in[k]}\sum_{x\in V_i}d^2(x,c_i)}{10k}.
\end{aligned}
\tag{6}
$$

Since there are at most $k-1$ such batches, we conclude that the their total contribution to the error $|\mathcal{K}_2(S,F,C)-\mathcal{K}_2(X,F,C)|$ can be upper bounded by

$$
\frac{\varepsilon\mathsf{OPT}}{5}+\frac{\varepsilon\mathcal{K}_2(X,F,C)}{10k}\leq \frac{\varepsilon}{3}\cdot\mathcal{K}_2(X,F,C).
$$

It completes the proof. $\qquad\qquad\qquad\qquad\qquad\qquad\qquad\qquad\qquad\qquad\qquad\qquad\quad\square$

## B.2 Extending to higher dimension

The extension is almost the same to fair $k$-median, except that we apply Theorem B.2 to construct the coreset on each line. Let $S$ denote the combination of coresets generated from all lines.

*Proof of Theorem B.1.* By the above construction, the coreset size is $O(k^3\varepsilon^{-d-1})$. For the correctness, Theorem 3.6 in [25] applies an important fact that for any $k$-subset $C\subseteq\mathbb{R}^d$,

$$
\mathcal{K}_2(X,C^\star)\leq c\cdot\mathcal{K}_2(X,C).
$$

In our setting, we have a similar property. Note that for any given `assignment constraint` $F\in\mathbb{R}^k$ and any $k$-subset $C\subseteq\mathbb{R}^d$, we have

$$
\mathcal{K}_2(X,C^\star)\leq c\cdot\mathcal{K}_2(X,F,C).
$$

585 Then combining this fact with Theorem B.2, we have that $S$ is an $\varepsilon$-coreset for the fair $k$-means
586 clustering problem, by the same argument as that of Theorem 3.6 in [25]. □

## B.3 Proof of Theorem 4.3

588 *Proof.* The proof idea is similar to that of Lemma 2.8 in [25]. We first rotate space such that the line
589 is on the $x$-axis and assume that $x_1 \le x_2 \le \ldots \le x_n$. Given an assignment constraint $F \in \mathbb{R}^k$ and a
590 $k$-subset $C = \{c_1, \ldots, c_k\} \subseteq \mathbb{R}^d$, let $c_i'$ denote the projection of point $c_i$ to the real line and assume
591 that $c_1' \le c_2' \le \ldots \le c_k'$. Our goal is to prove that

$$|\mathcal{K}_1(S, F, C) - \mathcal{K}_1(X, F, C)| \le \frac{\varepsilon}{3} \cdot \mathcal{K}_1(X, F, C).$$

592 By the construction of $S$, we build up a mapping $\pi : X \to S$ by letting $\pi(x) = \overline{B}$ for any $x \in B$.
593 For each $i \in [k]$, let $C_i$ denote the collection of points assigned to $c_i$ in the optimal fair $k$-median
594 clustering of $X$. By Lemma 4.1, $C_1, \ldots, C_k$ partition the line into at most $2k - 1$ intervals $\mathcal{I}_1, \ldots, \mathcal{I}_t$
595 ($t \le 2k - 1$), such that all points of any interval $\mathcal{I}_i$ are assigned to the same center. Denote an
596 assignment function $f : X \to C$ by $f(x) = c_i$ if $x \in C_i$. Let $\widehat{\mathcal{B}}$ denote the set of all batches $B$,
597 which intersects with more than one intervals $\mathcal{I}_i$, or alternatively, the interval $\mathcal{I}(B)$ contains the
598 projection of a center point of $C$ to the $x$-axis. Clearly, $|\widehat{\mathcal{B}}| \le 2k - 2 + k = 3k - 2$. For each batch
599 $B \in \widehat{\mathcal{B}}$, we have

$$\sum_{x \in B} d(\pi(x), f(x)) - d(x, f(x)) \stackrel{\text{triangle ineq.}}{\le} \sum_{x \in B} |d(x, \pi(x))| = \sum_{x \in B} |d(x, \overline{B})| \stackrel{\text{Defn. of } B}{\le} \frac{\varepsilon\mathsf{OPT}}{30k}. \quad (7)$$

600 Note that $X \setminus \bigcup_{B \in \widehat{\mathcal{B}}} B$ can be partitioned into at most $3k - 1$ contiguous intervals. Denote these
601 intervals by $\mathcal{I}_1', \ldots, \mathcal{I}_{t'}'$ ($t' \le 3k - 1$). By definition, all points of each interval $\mathcal{I}_i'$ are assigned to the
602 same center whose projection is outside $\mathcal{I}_i'$. Then by the proof of Lemma 2.8 in [25], we have that for
603 each $\mathcal{I}_i'$,

$$\sum_{x \in \mathcal{I}_i'} d(\pi(x), f(x)) - d(x, f(x)) \le 2\xi = \frac{\varepsilon\mathsf{OPT}}{15k}. \quad (8)$$

604 Combining Inequalities (7) and (8), we have

$$\mathcal{K}_1(S, F, C) - \mathcal{K}_1(X, F, C) \le \sum_{x \in X} d(\pi(x), f(x)) - d(x, f(x)) \quad (\text{Defn. of } \mathcal{K}_1(S, F, C))$$

$$= \sum_{B \in \widehat{\mathcal{B}}} \sum_{x \in B} d(\pi(x), f(x)) - d(x, f(x))$$

$$+ \sum_{i \in [t]} \sum_{x \in \mathcal{I}_i'} d(\pi(x), f(x)) - d(x, f(x)) \quad (9)$$

$$\le (3k - 2) \cdot \frac{\varepsilon\mathsf{OPT}}{30k} + (3k - 1) \cdot \frac{\varepsilon\mathsf{OPT}}{15k} \quad (\text{Ineqs. (7) and (8)})$$

$$\le \frac{\varepsilon\mathsf{OPT}}{3} \le \frac{\varepsilon}{3} \cdot \mathcal{K}_1(X, F, C).$$

605 To prove the other direction, we can regard $S$ as a collection of $n$ unweighted points and consider the
606 optimal fair $k$-median clustering of $S$. Again, the optimal fair $k$-median clustering of $S$ partitions
607 the $x$-axis into at most $2k - 1$ contiguous intervals, and can be described by an assignment function
608 $f' : S \to C$. Then we can build up a mapping $\pi' : S \to X$ as the inverse function of $\pi$. For each
609 batch $B$, let $S_B$ denote the collection of $|B|$ unweighted points located at $\overline{B}$. We have the following
610 inequality that is similar to Inequality (7)

$$\sum_{x \in S_B} d(\pi'(x), f'(x)) - d(x, f'(x)) \le \frac{\varepsilon\mathsf{OPT}}{30k}.$$

611 Suppose a contiguous interval $\mathcal{I}$ consists of several batches and satisfies that all points of $\mathcal{I} \cap S$ are
612 assigned to the same center by $f'$ whose projection is outside $\mathcal{I}$. Then by the proof of Lemma 2.8
613 in [25], we have that

$$\sum_{B \in \mathcal{I}} \sum_{x \in S_B} d(\pi'(x), f'(x)) - d(x, f'(x)) \le 0.$$

614     Then by a similar argument as for Inequality (9), we can prove the other direction

$$\mathcal{K}_1(X, F, C) - \mathcal{K}_1(S, F, C) \leq \frac{\varepsilon}{3} \cdot \mathcal{K}_1(X, F, C),$$

615     which completes the proof.         □

## Footnotes

[3]ID means that the function $f$ first (non-strictly) increases and then (non-strictly) decreases. DI means the other way round.

[4]For example, we can set $c = 10$ by [29].

[5]An $\varepsilon$-net $Q$ means that for any point $p$ in the unit sphere, there exists a point $q \in Q$ satisfying that $d(p, q) \leq \varepsilon$.

[6]Recall that a weighted point can be partially assigned to more than one cluster.