[Reviews · NeurIPS 2019]

Reviewer 1



This paper introduces a new coreset construction mechanism for fair clustering in which the points can be of multiple disjoint types. As in classic fair clustering, the goal of this work is to construct a clustering in which the types represented in each cluster are balanced. Unlike previous work, the focus here is on constructing the clustering efficiently via coresets. This work provides a coreset construction algorithm for fair k-median (previously unknown) and improves the previously known coreset construction algorithm for fair k-means. In addition to theoretical contributions with respect to coreset size and construction time, the authors also provide a small empirical study. The main strength of this paper is theoretical. The authors prove that their algorithms construct coresets with size independent of N for fair k-medians and k-means. To do so, they adapt the proof technique from a previous paper that does not deal with fair clustering to the fair clustering setting. This is a meaningful contribution as well as timely as fair clustering is an active and quickly growing research area. The primary weakness of this paper is the clarity of the technical sections. Section 3 helps with this, but Sections 4 and 5 are difficult to understand without carefully studying reference [25] (in fact, there is a longer version of Section 5 in the appendix!). Elements including an algorithm box and/or visual description of the 1-d case (section 4.1) would improve the presentation. The source code was helpful here although an algorithm box may be better for readers without access to the source. In the experiments, this paper compares their coreset construction algorithm to one other coreset construction algorithm for fair k-means. While this comparison is informative, the work misses other clear comparisons that would improve the paper. For example, there are no comparisons to fair clustering approaches that do not use coresets (for example, reference [4] in particular) to help explore the tradeoff between error and speed. **Edit**: thank you for including the additional experimentation. I think this does improve your paper. Additionally, your commitment to include visualization and pseudocode should improve readability. Originality: the paper exhibits a reasonable amount of originality. Many of the proof techniques and other ideas stem from reference 25. However, the authors show how that work can be extended to the fair clustering setting with multiple disjoint types, which requires novel analysis. Clarity: the writing and overall storyline of the paper is relatively clear however there are many details and definitions that are omitted that make understanding the paper challenging. For example, the definitions/differences between the terms “groups” and “types” is never made explicit. Also, this paper draws heavily on reference [25]; understanding the details this work was challenging for me without familiarity with [25]. Significance: this paper contributes to an active and important area of coresets for fair clustering problems. This includes both fair k-medians and fair k-means. The results presented in this paper will be important for other researchers studying scalable approaches to fair clustering with coresets. Quality: the theoretical pieces of this paper were challenging to evaluate but seem to be correct. === Smaller issues === Section 1: “Due the scale at which one is required to clustering”, ungrammatical Section 1: “Save the storage” → “Save storage” Section 1: “size independent on N” → “size independent of N” Section 1: “with size depend on” → “with size that depends on” Definition 2.1: K_z(S,F,C) \in (1 \pm \epsilon) K_z(X,F,C). I don’t think that “\in” is the correct notation here since K_z is a value and not a set. Section 4: “In a high level” → “at a high level” Section 4: “is the same to” → “is the same as” Section 5: “we show how the construction of coresets” → “we show how to construct coresets” Section 5: “same to [25]” → “similar to [25]” General comment: I think the terms “collections of groups” and “types” in this paper are closely connected but never defined and thus a bit confusing. Define (roughly) these and give examples in the introduction. Later on this becomes more clear.

Reviewer 2



(Originality) The authors claims the novelty lies with the construction of a novel epsilon-corsets with k-median and k-mean objective. The constructed corsets which is of smaller size than the full set is used in existing fair clustering algorithms. This construction is very similar to that proposed in [25]. Originality seems minor. (Quality) It would be more meaningful if the corset construction was conducted with k-center objective. (Clarity) Well written and relatively clear- better if there were illustrations. (significance) Empirical error that is superlinear with increasing epsilon. How is this a concern?

Reviewer 3



The paper clearly identifies which parts/techniques are taken from other papers (i.e. known coreset construction from Har-Peled et al), and where extensions have been made. It also gives a good overivew on the contributions and experiments and is easy to follow. The main techniques behind the theoretical coreset constructions are well-known from [25]. The two new components are the observation regarding the reduction of constraint types (theorem 4.2) and the Lemmata regarding the analysis of the number of "contiguous intervals" (Lemma 4.1 and Lemma 5.1 / B.2). (Note that Theorem 4.2 is a generalisation of Lemma 6 in [36].) It is great that, in contrast to many theoretical clustering papers, this paper includes an evaluation of a practical version of the algorithm that's analysed in the theoretical part of the paper. However, there are also some things about the experiments that I struggle with: * As always, it is a bit hard to interpret the cost alone (without any goal of what actually will be done with the clustering). For errors of up to 10%, the results seem to be comparable with the BICO solution [36]. After that, there's a small gap (up to about 5%) between the solutions. I'm not sure if this is significant or not. * There is *no* comparison to simpler baselines, ie.g. to uniform sampling where, to ensure fairness, one could just sample uniformly at random per "constraint type". (One might suspect that the number of constraint types grows, the performance of this simple baseline should become better and better (as the constraint types pre-partition the space kind of)). * The experiments are only on two specific data sets. Given the fast processing times (cf. T_S and T_X in Tables 3 and 4), it is surprising that not more datasets have been considered. * ... and the times needed to compute the coresets are not reported. Moreover, there is one point (right in the problem definition section) that remains unclear to me: the re-definition described in ll. 131 (after Definition 2.1): "a point [...] may be partially assigned to more than one cluster". This seems unnecessary. I guess the original intent was just to point out that the pointwise costs are now weighted? (otherwise, I might be missing something here) Besides, some minor remarks: - Theorem 4.2 is not a pure "existence" statement (and it would be useless in the following if it was just that) - this should be made clear. - Table 2 is a bit confusing since some rows still contain eps, which should be Omega(1) - It should be made clear that a practial variant of the theoretical algorithm is evaluated. - There are some typos: several upper/lower-case typos in the references , l. 40 (due to the scale at which one is required to clustering), l. 175 (let ... denotes the)

[Author Response · NeurIPS 2019]

**Response for #4148, "Coresets for Clustering with Fairness Constraints"**
We thank the reviewers for their valuable comments. We start with answering questions raised by more than one
reviewer and then answer reviewer-specific questions.

**New experiments (Reviewers 1 and 3):** Based on your suggestions, we conducted new experiments to study the
speed-up our coresets obtain for recent fair clustering algorithms [3, 4]. We observed that simple adaption of our
coresets already offers at least a 10x speed-up to **FairTree** [3] and a 2x speed-up to **FairLP** [4], even taking the coreset
construction time into consideration; see Table 1a for some preliminary results. (The same coreset could be used for
clustering with any fairness constraints, so its construction time would be averaged out if multiple fair clustering tasks
are performed.) We believe that with a carefully crafted implementation of these results that integrate with our coresets
would further accelerate the running time. We will include a thorough comparison in the final version of the paper.

Table 1: new empirical results

| | $T_{\mathrm{ALG}}$ (s) | $T'_{\mathrm{ALG}}$ (s) | $T_{\mathrm{COR}}$ (s) |
|---|---|---|---|
| **FairTree** | 444.31 | 22.70 | 26.14 |
| **FairLP** | 425.70 | 169.05 | - |

| $\epsilon$ | Ours | emp. err. BICO | Uni. | size | time (ms) |
|---|---|---|---|---|---|
| 10% | 0.28% | 1.04% | 10.63% | 880 | 43.88 |
| 20% | 0.55% | 1.12% | 28.73% | 610 | 29.36 |
| 30% | 1.17% | 4.06% | 19.91% | 503 | 25.97 |
| 40% | 2.40% | 4.45% | 48.10% | 433 | 22.17 |

(a) speed-up of fair clustering using coresets on **Census1990** (n $\approx$ 2.5m). $T_{\mathrm{ALG}}$ is run time w/o coreset, $T'_{\mathrm{ALG}}$ is run time on top of our coreset, and $T_{\mathrm{COR}}$ is time to construct coreset.

(b) performance of $\varepsilon$-coresets for fair $k$-means w.r.t. varying $\varepsilon$ on **Adult** (n $\approx$ 30k), with gender and marital status as sensitive attributes.

We also conducted experiments on additional datasets including **Athletes** and **Diabetes**, and we employed a new
uniform sampling baseline. We report partial results on **Adult** dataset in Table 1b, and complete results will appear
in the final version of the paper. The results in the table show that the empirical error of our coreset is at most $50\%$
of **BICO**, and $10\%$ of **Uniform**. We note that the performance improvement to **BICO** is more significant than in our
submission version due to the recently improved implementation of our coresets. Our coreset construction time scales
roughly linearly with the size of the coreset, and as mentioned above, its efficiency leads to accelerated fair clustering
algorithms (Table 1a). We also observe that the performance of **Uniform** is comparable to ours and **BICO** on certain
datasets such as **Diabetes**, but its worst-case error is unbounded in general.

**Novel contributions (Reviewers 2 and 3):** Conceptually, we obtain the first coresets for fair clustering w.r.t. *multiple*
*types* whose size is *independent* of the size of the dataset. Technically, to handle the multiple types, we show a general
reduction that turns any fair coreset for a binary type into a coreset for multiple types. Although our coreset for a binary
type is based on the framework of [25], [25] does not readily apply to fair clustering, due to the technical hurdle that
data points may not be assigned to the nearest center in an optimal fair clustering. We cross this hurdle by showing new
structural lemmas (Lemmas 4.1 and B.2) for 1-dimensional dataset $X$, that the optimal fair $k$-median/means clustering
partitions $X$ into $O(k)$ contiguous intervals. Empirically, as in Table 1a, our coreset may be applied to speed-up several
very recent fair clustering algorithms [3,4] on various datasets, achieving more than 10x acceleration for [3].

**Presentation (Reviewers 1 and 2):** In the final version, we will add pseudocodes of our algorithms, a figure to visualize
the 1D case in Section 4.1, and examples to illustrate the coreset construction. We will also make the paper more
self-contained by adding a short "background" section that explains the framework of [25] and its technical limitations.

**Reviewer 1:** Regarding the terms "collections of groups" and "types": We will clarify their definitions in the final
version of the paper.

**Reviewer 2:** Regarding the $k$-center objective: Our coreset construction does work for $k$-center by combining Theorem
4.2 and the results of [24], which was also mentioned in Footnote 1.

Regarding the concern that the empirical error is superlinear with increasing $\varepsilon$: We didn't quite understand your
question. The only part that the empirical error seems to be superlinear is in Table 1 (Bank) for $\varepsilon$ between $30\%$ and
$40\%$. Otherwise, the empirical error is roughly linear with increasing $\varepsilon$.

**Reviewer 3:** Regarding the re-definition that a weighted point may be partially assigned to more than one cluster: We
do need the re-definition. The reason is that if each weighted point can only be assigned to one cluster, it is possible
that the assignment constraint cannot be satisfied. For instance, suppose a coreset only contains one point with weight
100 and we require that each of two centers should be assigned by exactly 50 points. Then this point must be partially
assigned to two centers.

Regarding the comment on Theorem 4.2: Theorem 4.2 actually implies an algorithm that constructs a coreset w.r.t.
multiple types. We will clarify this and also discuss the running time in the final version of the paper.

[Meta-Review · NeurIPS 2019]

The main strength of the paper as seen by the reviewers is a) the theoretical development of core sets and b) the elaboration of empirical work (also in the response). The "modularity" of the approach is also attractive as is the potential for scalable solutions. The weakness of the paper is its lack of generality (it can only address one kind of fairness definition). OVerall it's a reasonable work -- it could do with more work on presentation and visualization but the reviewers deemed it satisfactory for acceptance.